# Transcriptome Analysis and Identification of Genes Associated with Starch Metabolism in *Castanea henryi* Seed (Fagaceae)

**DOI:** 10.3390/ijms21041431

**Published:** 2020-02-20

**Authors:** Bin Liu, Ruqiang Lin, Yuting Jiang, Shuzhen Jiang, Yuanfang Xiong, Hui Lian, Qinmeng Zeng, Xuedie Liu, Zhong-Jian Liu, Shipin Chen

**Affiliations:** 1College of Forestry, Fujian Agriculture and Forestry University, Fuzhou 350002, China; 1170428002@fafu.deu.cn (B.L.); 1180428009@fafu.edu.cn (R.L.); fafujiangyuting@163.com (Y.J.); 1190428006@fafu.edu.cn (S.J.); 1190455005@fafu.edu.cn (Y.X.); 1190455001@fafu.edu.cn (H.L.); fafuzqm@163.com (Q.Z.); 1180428010@fafu.edu.cn (X.L.); 2Key Laboratory of National Forestry and Grassland Administration for Orchid Conservation and Utilization at College of Landscape, Fujian Agriculture and Forestry University, Fuzhou 350002, China; zjliu@fafu.edu.cn

**Keywords:** *Castanea henryi*, seed germination, starch, transcriptome analysis

## Abstract

Starch is the most important form of carbohydrate storage and is the major energy reserve in some seeds, especially *Castanea henryi*. Seed germination is the beginning of the plant’s life cycle, and starch metabolism is important for seed germination. As a complex metabolic pathway, the regulation of starch metabolism in *C. henryi* is still poorly understood. To explore the mechanism of starch metabolism during the germination of *C. henryi*, we conducted a comparative gene expression analysis at the transcriptional level using RNA-seq across four different germination stages, and analyzed the changes in the starch and soluble sugar contents. The results showed that the starch content increased in 0–10 days and decreased in 10–35 days, while the soluble sugar content continuously decreased in 0–30 days and increased in 30–35 days. We identified 49 candidate genes that may be associated with starch and sucrose metabolism. Three ADP-glucose pyrophosphorylase (AGPase) genes, two nucleotide pyrophosphatase/phosphodiesterases (NPPS) genes and three starch synthases (SS) genes may be related to starch accumulation. Quantitative real-time polymerase chain reaction (qRT-PCR) was used to validate the expression levels of these genes. Our study combined transcriptome data with physiological and biochemical data, revealing potential candidate genes that affect starch metabolism during seed germination, and provides important data about starch metabolism and seed germination in seed plants.

## 1. Introduction

Seed germination is the basis of plant formation and is the period in the life of a plant with the highest metabolic activity. It is related to the subsequent growth and final yield of plants. Seeds are a vital component of diets worldwide, and seed biology is one of the most extensively researched areas in plant physiology. During the period of germination, the metabolism of fats, proteins and carbohydrates provide energy for seedling growth [1,2,3]. Carbohydrates are an important source of cellular energy and are closely related to seed vigor and the germination process [4]. Starch is the most important carbohydrate reserves in plants and is mainly composed of amylopectin and amylose [5,6]. Amylose is essentially a linear molecule in which glucosyl monomers are joined via α-1,4 linkages, and amylopectin is composed of linked tandem clusters in which linear α-1,4-glucan chains are regularly branched via α-1,6-glucosidic linkages [7,8,9].

Seed starch is a primary source of carbohydrate for both human and animal diets. Seed starch is the major storage compound that accumulates in the cereal endosperm, and more than half of the world’s population uses rice as a source of carbon intake every day [10]. Starch and sucrose are metabolized in plants through the amount of complex and coordinated pathways that are regulated by multiple enzymes [6,11]. The key enzymes include ADP-glucose pyrophosphorylase (AGPase, EC 2.7.7.27), starch synthases/granule-bound starch synthases (SS/GBSS, EC 2.4.1.21), starch-branching enzyme (SBE, EC 2.4.1.18), starch debranching enzyme (DBE, EC3.2.1.10), isoamylase (ISA, EC3.2.1.68), alpha-amylase (AMY, EC 3.2.1.1), beta-amylase (BMY, EC 3.2.1.2), alpha-glucosidase (MAL EC3.2.1.20), starch phosphorylase (SP, EC2.4.1.1), sucrose synthase (SuSy, EC 2.4.1.13), sucrose-phosphate synthase (SPS, EC 2.4.1.14) and invertase (INV, EC 3.2.1.26) [12,13,14,15,16,17].

Sucrose is hydrolyzed by sucrose synthases (SuSy) and invertases (INVs) into monomer sugars, glucose/UDP-glucose (UGP2) and fructose, which are then converted through a series of reactions into glucose-1-phosphate, which is considered to be the most efficient starch synthesis precursor [18]. Starch synthesis begins with the formation of an activated glucosyl donor, ADP-glucose (ADPG), in a reaction catalyzed by ADPG pyrophosphorylase. The enzymatic reactions that utilize ADPG to build α-(1→4)-linked “linear” chains are catalyzed by the starch synthases (SS). Branch linkages are introduced by branching enzymes (SBE) that catalyze the cleavage of an internal α-(1→4) linkage and transfer the released reducing end to a C6 hydroxyl, creating a new α-(1→6) linkage [19]. Amylose is synthesized by ADP-Glc pyrophosphorylase (AGPase) and granule-bound starch synthase (GBSS), while amylopectin is synthesized by the coordinated actions of AGPase, soluble starch synthase (SS), SBE, and the starch debranching enzyme (DBE) [20,21].

Candidate genes associated with starch biosynthesis and metabolism have been widely studied in many plants [22]. However, the key genes and how their expression levels are regulated for in starch biosynthesis differ among species. In the northern hemisphere, *Fagaceae* are the main tree species of tropical, subtropical and temperate forests, and they are also the most significant dominant species of angiosperms. Many of the *Fagaceae* plants are important economic plants and dual-purpose plants [23,24]. *Castanea henryi* is a member of the *Castanea* genus of the *Fagaceae* family. *Castanea* trees are extensively distributed between eastern Asia and eastern North America and are used to produce wood, tanning agents and food [25,26].

Chestnut fruits have considerable economic value. Chestnuts are rich in nutrients and contain starch, soluble sugar, protein, lipids and 18 types of amino acids, including eight that are essential to the human body. The nutritional value of chestnut fruits is higher than flour, rice and potato [27,28,29]. These fruits have a unique flavor and taste and can be used as a staple food similar to potatoes or cereals [30]. In addition to being a popular dried fruit, chestnut fruits have long been used as a traditional Chinese medicine [31]. *Fagaceae* plants play an important role in the forest ecosystem; however, previous studies have not focused on the germination of *Fagaceae* seeds [27,32]. In the present study, we sequenced the transcriptome of *C. henryi* seeds at four germination stages to identify the candidate genes involved in starch and sucrose metabolism. The expression profiles of a few genes were further validated by quantitative real-time polymerase chain reaction (qRT-PCR). The results presented herein may be useful for characterizing the molecular mechanism underlying starch and sucrose metabolism in seed germination.

## 2. Results

### 2.1. Morphological Evaluation of the Seed and Starch and Sugar Analysis

Seeds were sampled from 0 days after sowing (DAS) to 35 DAS, approximately every 5 days, throughout the germination of the seeds. The morphological changes in the *C. henryi* seeds were obvious (Figure 1). The white radicle extended downward at 10 days after sowing; the surface layer of the radicle was brown and deepened, with a large number of lateral roots growing, and the germ was formed between 10 and 20 DAS. The lateral roots gradually grew stronger, and the buds extended to form stems and leaves between 20 and 30 DAS, with 2–5 young leaves spreading by the end of this period.

To investigate the changes in starch and sucrose metabolism in the seeds, the starch and soluble sugar contents of the dry matter were measured. The starch content increased from 50.60% to 60.00% from day 0 to 10 DAS, then slowly decreased from 60.00% to 53.07% from 10 DAS to 30 DAS, followed by rapidly decreasing to 38.83% (Figure 2). The soluble sugar rapidly decreased from 19.81% to 10.10% from 0 DAS to 10 DAS, and then slowly decreased from 10.10% to 8.24% from 10 DAS to 30 DAS, followed by rapidly increasing to 11.41% (Figure 3).

Based on these results, four different periods, T01, T02, T03 and T04 (0, 10, 20 and 30 DAS), were selected for comparative transcriptome analysis to better explore the molecular and metabolic regulatory mechanisms of starch and sucrose metabolism during the germination of *C. henryi* seeds.

### 2.2. Overview of Transcriptome Sequencing

In total, twelve cDNA libraries with three repetitions for each stage were constructed and sequenced across the stages of seed germination: 70.19, 67.68 and 70.19 million raw reads were generated for T01 stage; 67.68, 70.19 and 70.19 million raw reads were generated for T02 stage; 70.19, 67.69 and 70.19 million raw reads were generated for T03 stage; and 70.19, 65.7 and 67.68 million raw reads were generated for T04 stage. After removing the adaptor sequences, low-quality and high content of unknown base N reads, the clean rates were all higher than 92.28%, and the Q20 values were all over 92%. Thus, a total of 76.903 Gb-cleaned reads (81.25%) were mapped to the *C. henryi* genome, and 52.94% of the mapped reads were unique to the *C. henryi* genome. An overview of the sequencing statistics is shown in Appendix A. The length distribution of the *C. henryi* genes is shown in Appendix A. The length of the assembled genes ranged from 300 to 1000 nt and accounted for 48.12% of all transcripts. In addition, 17,283 transcripts (51.88%) had lengths longer than 1 kb. All the clean reads were subsequently subjected to de novo assembly with the StringTie, Cufflinks and CPC programs, resulting in 33,314 transcripts (Appendix A).

### 2.3. Functional Annotation of Genes and Co-Expression Analysis

Among the 36,734 genes, 32,766 (89.20%) could be annotated, and 100 genes could be matched with all of the databases (Appendix A and Appendix A). In particular, 1391 (3.79%), 1880 (5.12%), 15,275 (41.58%), 26,437 (71.97%) and 30,041 (81.78%) genes were aligned to the TF (PlantTFDB), PRG (Plant Resistance Gene Database, PRGdb), GO (Gene ontology), KEGG (Kyoto Encyclopedia of Genes and Genomes) and Nr (NCBI non-redundant protein sequences) protein databases, respectively.

To assign the genes capable of encoding transcription factors (TF), 1391 genes were annotated to 59 TF families (Appendix A). Among these TF families, MYB (207) was the most abundant, followed by AP2-EREBP (132), NAC (101), bHLH (100), WRKY (58), C2H2 (54) and ABI3VP1 (53).

A total of 15,275 (41.58%) genes had significant hits in the GO database, which further classified them into 47 terms: biological process (20 terms), cellular component (15 terms), and molecular function (12 terms) (Appendix A). In particular, ‘cellular process’ and ‘cell’ were the most enriched components in ‘biological process’ and ‘cellular component’, and ‘catalytic activity’ and ‘binding’ were the most enriched components in the ‘molecular function’ category. Among the three main categories, only nine genes were assigned to ‘biological adhesion’ (two), ‘protein tag’ (two), and ‘nucleoid’ (five).

To identify the active biological pathways in seed germination, the assembled genes were mapped to the KEGG protein database. Genes were classified into 21 terms: metabolism (11 terms), genetic information processing (four terms), environmental information processing (two terms), organismal systems (one term), cellular processes (one term) and human diseases (two terms) (Appendix A). In particular, ‘global and overview maps’, ‘carbohydrate metabolism’, ‘translation’, ‘folding, sorting and degradation’, and ‘environmental adaptation’ represented the top five largest enriched KEGG pathways.

### 2.4. Identification and Selection of DEGs

The DEGs were analyzed to identify candidate genes related to starch and sucrose metabolism. The expression levels of genes involved in *C. henryi* seed germination were explored with the DEGseq Package [33]. After being filtered (Fold Change ≥ 2 and Adjusted *p*-value ≤ 0.001), 12,943, 11,999, 13,103, 5641, 5952 and 6206 were searched through the transcriptome analysis and comparisons were made between T01 and T02 stage, T01 and T03 stage, T01 and T04 stage, T02 and T03 stage, T02 and T04 stage, and T03 and T04 stage, respectively. The T02/T03 comparison produced the fewest DEGs (2222 genes upregulated and 3491 genes downregulated), while the T01/T04 comparison produced the most DEGs (6748 genes upregulated and 6355 genes downregulated) (Figure 4). Moreover, volcano plots were constructed to identify the transcripts that significantly changed during the germination process, and the significant DEGs are represented with red dots (Figure 5).

### 2.5. Identification of Genes Critical for Starch and Sucrose Metabolism in C. henryi Seeds

The starch and sucrose metabolism pathways are well characterized, and many critical genes have been identified [34,35,36,37]. The genes encoding the key enzymes involved in starch and sucrose metabolism showed different expression patterns during the germination of the seeds (Figure 6 and Appendix A).

In the first step of starch biosynthesis, AGPase (EC 2.7.7.27) catalyzes the conversion of glucose-1-phosphate into ADP-glucose [6]. We identified four AGPase homologs in our annotated *C. henryi* transcriptome gene dataset, two of which were highly expressed with FPKM values > 100 (Che007374 and Che013145) and two of which were significantly decreased in T01 to T03 (Che003775 and Che013145).

Nucleotide pyrophosphatase/phosphodiesterases (NPPs, EC 3.6.1.9) are widely distributed N-glycosylated enzymes that catalyze the hydrolytic breakdown of the pyrophosphate and phosphodiester bonds of numerous nucleotides and nucleotide sugars, and previous studies have suggested that NPP1 exhibits the highest hydrolyzing activity towards ADP-glucose, ADP-ribose, ATP and UDP-glucose [38,39,40,41]. In our results, three NPPS-encoding genes (Che006880, Che018517, and Che032493) were differently expressed during seed germination, and two genes (Che006880 and Che032493) were expressed at higher levels in T01.

Additionally, two GBSS genes were predicted in our dataset; one gene was highly expressed with an FPKM value > 3000 (Che029279). Eight SS genes and five SBE genes were also predicted, and four of the eight SS genes were highly expressed (Che013592, Che022142, Che022683, and Che036389) in T01 and similarly expressed in T02 to T04. One SBE gene was highly expressed and continued to increase from T01 to T04. Starch synthases (SS, EC 2.4.1.21) utilize ADPG to elongate linear chains by catalysing the formation of new α-(1→4) linkages. One particular SS, granule-bound starch synthase (GBSS, EC 2.4.1.424), is responsible for the production of the long linear chains in the amylose component of starch. Starch branching enzymes (SBE, EC 2.4.1.18) introduce α-(1→6) branch linkages by cleaving an internal α-(1→4) bond within a linear chain and transferring the released reducing end to a C6 hydroxyl [19].

Starch is degraded by alpha-amylase (AMY, EC 3.2.1.1), beta-amylase (BMY, EC 3.2.1.2) or via the phosphorylation pathway. Starch phosphorylase (SP, EC 2.4.1.1) catalyzes starch degradation via the reversible phosphorylation of α-glucan [5]. In this study, thirty-seven genes encoding AMY, sixteen genes encoding BMY and eight genes encoding SP were identified in our sequencing data. Furthermore, two ISA genes were identified, with Che019232 being the most abundantly expressed. Isoamylase (ISA, EC3.2.1.68) hydrolyzes the α-(l→6) branch linkages of starch. ISA has the specificity to hydrolyze the α-(l→6) branch linkages of relatively long starch chains and can completely hydrolyze all of the branch linkages of amylopectin [42,43]. The enzyme is reported to be similar to glucoamylase and has a starch-binding domain [44]. The other important enzyme is alpha-glucosidase or maltase (EC 3.2.1.20), which is mainly involved in the metabolism of starch and its derivatives. This enzyme catalyzes the final step of the digestive process of carbohydrates, mainly starch, by acting on maltose and other short malto-oligosaccharides produced by amylases cleaving 1,4-alpha bonds and producing glucose as the final product [45,46]. Nineteen MAL genes were identified, four that were expressed significantly higher in T02–T04 than in T01.

Sucrose synthase (SuSy, EC 2.4.1.13) mainly decomposes sucrose into UDP-glucose and fructose to provide substrates for starch synthesis and catalyzes the reversible conversion of UDP-glucose and fructose into sucrose [5,47]. Ten SuSy genes were identified in our dataset, with Chen028629 being the most abundantly expressed. UDP-glucose pyrophosphorylase (UGPase, EC 2.7.7.9) constitutes a reversible enzymatic step for interconversions between starch and sucrose metabolites and is responsible for the synthesis and metabolism of UDP-glucose [48]. Six UGPase genes were identified in our dataset, and the fact that Che002934 was the most highly expressed suggests it may be crucial for the conversion between starch and sucrose metabolites. Sucrose–phosphate synthase (SPS, EC 2.4.1.14) converts UDP-glucose and fructose-6-phosphate into sucrose-6-phosphate, and sucrose–phosphate phosphatase (SPP, EC 3.1.3.24) converts sucrose-6-phosphate into sucrose [49]. Four SPS genes and three SPP genes were predicted in our dataset, and one SPP gene was highly expressed (Che022264). Sucrose is hydrolyzed to glucose and fructose by invertase (INV, EC 3.2.1.26) [50]. Three SuSy genes and two SPS genes were predicted and annotated in the present study, and most of them exhibited low expression. A large number of INV (57) genes were identified in our dataset, but almost all were only expressed at low levels in all four analyzed stages.

### 2.6. qRT-PCR Gene Expression Analysis

To verify the reliability of the RNA-seq data, seven candidate genes with differentially expressed transcripts involved in starch and sucrose metabolism were selected for real-time quantitative polymerase chain reaction (qRT-PCR) analysis (Appendix A). The qRT-PCR results were generally consistent with the high-throughput sequencing results (Figure 7). The levels of the transcripts decreased during the germination process, while the transcript of Che006880 (NPPS) decreased significantly in T01 to T02, gradually increased in T02 to T03, and then gradually decreased in T03 to T04. Moreover, the expression levels of Che032493 (NPPS), Che013592 (SS), Che022683 (SS), and Che036389 (SS) were all highest in T01, decreased significantly in T01–T02, and increased slightly in T02–T04. Furthermore, the expression levels of Che003775 (AGPase) and Che013145 (AGPase) were significantly decreased in T01 to T03, but then the expression of Che00377 (AGPase) slightly increased, and the expression of Che013145 (AGPase) significantly increased later. The fact that the qRT-PCR data were generally consistent with the high-throughput results suggests that the high-throughput sequencing data are reliable.

## 3. Discussion

*Castanea* trees are found all over the world, and their chestnut fruits are rich in nutritional elements and have considerable economic value [26]. The chestnut fruits produce seeds, and seed germination is one of the most important phases in the life cycle of a plant. A large number of nutrients are accumulated during the process of fruit formation, which are mainly used to prepare for seed germination. However, previous studies have mainly focused on the composition of the raw fruits and the process of material accumulation during seed development [36,51], and less is known about the mechanism of seed germination. In addition, starch and sucrose metabolism in *C. henryi* during germination is not fully understood, and full elucidation of the expression levels of genes involved in starch metabolism during germination has not been reported yet.

During the germination, the quiescent dry seed rapidly resumes metabolic activity upon imbibition, and the first change is the resumption of respiratory activity. After a steep initial increase in oxygen consumption, the rate declines until the radicle penetrates the surrounding structures, when another burst of respiratory activity occurs [52,53]. The glycolytic and oxidative pentose phosphate pathways both resume during phase I, and the Krebs cycle enzymes become activated [54,55]. In the early stage, the efficiency of the Krebs cycle is low, and the energy supply mainly depends on the glycolytic and oxidative pentose phosphate pathway. Carbohydrates are an important source of cell energy; therefore, a good supply of carbohydrates is closely related to seed vigour and the germination process [4]. Sugar is the most important carbohydrate reserve in plants, and there are various forms of sugars in seeds, which can be divided into soluble sugar and insoluble sugar according to their solubility in water. Soluble sugar mainly includes glucose, fructose, sucrose, maltose, stachyose and raffinose, while insoluble sugar mainly includes starch [56]. In the process of seed germination, the content of soluble sugar has two trends: for starch seeds, such as corn [57] and sorghum [58], the content of soluble sugar increases during germination; for fat seeds, such as *Pinus tabulaeformis* [59], the sugar content first decreases and then increases. The content of starch mainly shows a decreasing trend, such as in *Oryza sativa*, *Oryza sativa*, *Sorghum bicolor*, *Linum usitatissimum*, *Medicago truncatula*, and *Arabidopsis thaliana* [60].

Interestingly, in this study, we measured the contents of starch and soluble sugar throughout germination, and the results showed that the changes in the starch and soluble sugar contents in *C. henryi* are different to those in most plants. The starch content increased from the T01 to T02 stages, and then began slowly decreasing, while the soluble sugar continued decreasing throughout the germination. The observed changes in the starch and soluble sugar content were consistent with the results of previous studies involving other plant species. For example, in Hildebrand’s study, the changes in starch and soluble sugar content for four different genotypes were similar in that the starch content increased in the early stage and continuously decreased in the later stage, while the soluble sugar content decreased throughout the germination [61]. In Guzman’s research, the total starch composition of rice varied significantly during germination, but the starch mobilization patterns of the five GI lines differed but showed two distinct patterns. Starch in the waxy and low amylose varieties (IR65, IR24) continued decreasing, while in the intermediate (IR64), high (IR36) and very high (IR36 and IR36ae) amylose types, starch levels increased during the first two days then began to steadily decrease [62]. Thus, the dynamic alteration of starch and soluble sugar during chestnut seed germination implies that starch and soluble sugar may play different roles in different stages. In the process of seed germination leading to the growth of the main root, soluble sugar provides the main energy, while in the process of fibrous root and epicotyl growth, starch and soluble sugar both provide energy. Our findings led us to realize that this pattern of changes in the starch and soluble sugar content during chestnut germination is an interesting phenomenon.

In the present study, we used RNA sequencing technology to completely analyze the chestnut seed transcriptome at four germination stages. A total of 15 categories and 49 functional genes in our transcriptome dataset were identified based on the known starch and sucrose metabolism pathway [18,63,64,65], and the 15 key enzymes were encoded by more than one annotated gene. The expression levels of the starch and sucrose metabolism key enzymes genes were strongly correlated with the changes in the starch and soluble sugar content in the chestnut seeds, as indicated by most genes related to synthesis showing a downward trend while most of the genes related to degradation showed an upward trend.

Starch biosynthesis is a complex system composed of multiple subunits or isoforms of four classes of enzymes: ADP-glucose pyrophosphorylase (AGPase), starch synthase/granule-bound starch synthase (SS/GBSS), and starch branching enzyme (SBE). Each enzyme plays a distinct role, but presumably functions as part of a network [21,65,66]. The NPPs are also important because they catalyse the hydrolytic UDP-glucose to produce precursors for the synthesis of starch [41]. Two AGPase genes, two NPPS genes, two SS genes, one GBSS and two SBE genes were highly expressed during germination. The GBSS genes were expressed at the highest levels, with an FPKM value >3000, which suggests that GBSS may be the key starch synthesis enzyme. GBSS is an important enzyme that is responsible for amylose synthesis. Plants might regulate amylose metabolism by affecting the levels of GBSS and the quantity or size of starch granules in banana fruit during development or storage [67,68]. Starch is degraded by amylase (α-amylase, β-amylase), starch phosphorylase (SP) and isoamylase (ISA) [5,42,43]. Three α-amylase genes, four β-amylase genes, three SP genes and one ISA gene were highly expressed during germination. The highest FPKM value of β-amylase gene expression was 1055, which indicates that starch degradation must be catalyzed by more than one enzyme and that β-amylase may be an important starch degrading enzyme. A study of a downregulated form of plastidial b-amylase in potato leaves, which showed reduced rates of starch degradation at night, was the first clear evidence for b-amylase having a key role in transitory starch degradation in leaves [69,70].

Sucrose is mainly metabolized in two ways: via invertase (INV) and via sucrose synthase (SUSY). INV catalyzes sucrose conversion into glucose and fructose in the apoplast and intracellularly, and sucrose synthase (SUSY) catalyzes the reversible conversion of UDP-glucose and fructose into sucrose. In addition, sucrose metabolism includes sucrose–phosphate synthase (SPS), sucrose–phosphatase (SPP) and UDP-glucose pyrophosphorylase (UGPase) [5,47,48,50]. In our results, the expression of one SUSY gene was much higher than that of the other genes, which indicates that SUSY plays the most important role in the process of sucrose metabolism. The expression of SUSY was the highest in the T02 period, and the maximum rate of the decrease in the soluble sugar content was observed in the T01–T02 period. SuSy catalyzes a reversible reaction in vivo; however, our results indicate that SuSy mainly decomposes sucrose into UDP-glucose and fructose. The enzyme is thought to function primarily in the pathway of sucrose cleavage in plant sink tissues supplied with ample sucrose [71].

## 4. Materials and Methods

### 4.1. Plant Materials

The seeds which were genetically related were germinated in a greenhouse at Fujian Agriculture and Forestry University in Fuzhou, China. A total of 30 seeds sown in six different flowerpots were sampled from 0 days after sowing (DAS) to 35 DAS, approximately every 5 days, between 11 and 12 am. Samples were flash frozen in liquid nitrogen and stored at −80 °C until use. In this study, four periods of seed sampling (0, 10, 20, and 30 DAS) were selected for transcriptome sequencing.

### 4.2. Measurement of Starch and Soluble Sugar Content

Sample preparation: first, we dried the seeds at 50 °C for 72 h to a constant weight and ground them, including radicles, etc. Next, we placed 50 mg into a 10 mL test tube, added 4 mL of 80% ethanol, stirred and heated it in a water bath for 40 min, and, after cooling, collected the supernatant by centrifugation. We added 2 mL of 80% ethanol to the residue, extracted it twice, combined all of the supernatants and then added 10 mg of active carbon. We decolorized this mixture at 80 °C for 30 min, and then diluted it to 10 mL with 80% ethanol for soluble sugar determination. After drying the residue, we transferred it into a 50 mL volumetric flask, added 20 mL of hot distilled water, heated it in a boiling water bath for 15 min, added 2 mL of 9.2 mol/L perchloric acid to extract it for 15 min, kept it at a constant volume after room temperature cooling, mixed it evenly, filtered it, and measured the starch content of the filtrate. The content of soluble sugar and starch was determined referencing the anthrone colorimetry method [72].

### 4.3. RNA Extraction and High-Throughput Sequencing

Based on the previous results of the physical indicators of and changes in soluble sugar and starch content, the samples from four periods (0, 10, 20 and 30 days after germination) were selected as materials for comparative transcriptome analysis, and three biological replicates were used in this work. An ethanol precipitation protocol and CTAB-PBIOZOL reagent were used for the purification of total RNA from the *C. henryi* tissue according to the manual instructions: grind tissue samples about 80 mg with liquid nitrogen into powder and transfer the powder samples in 1.5 mL of preheated 65 °C CTAB-pBIOZOL reagents. The samples were incubated by Thermo mixer for 15 min at 65 °C to permit the complete dissociation of nucleoprotein complexes. After centrifuge at 12,000× *g* for 5 min at 4 °C, the supernatant was added—400 μL of chloroform per 1.5 mL of CTAB-pBIOZOL reagent—and was centrifuged at 12,000× *g* for 10 min at 4 °C. The supernatant was transferred to a new 2.0 mL tube, then 700 μL acidic phenol and 200 μL chloroform were added, followed by centrifuging 12,000× *g* for 10 min at 4 °C. An equal volume of aqueous phase of chloroform was added and centrifuged at 12,000× *g* for 10 min at 4 °C. An equal volume of a supernatant of isopropyl alcohol was added and placed at −20 °C for 2 h for precipitation. After that, the mix was centrifuged at 12,000× *g* for 20 min at 4 °C and then the supernatant was removed. After being washed with 1 mL of 75% ethanol, the RNA pellet was air-dried in the biosafety cabinet and was dissolved by adding 50 µL of DEPC-treated water.

Subsequently, the total RNA was qualified and quantified using a Nano Drop and Agilent 2100 Bioanalyzer (Thermo Fisher Scientific, MA, USA). Oligo(dT)-attached magnetic beads were used to purified the mRNA. The purified mRNA was fragmented into small pieces with the fragment buffer at the appropriate temperature. Then, first-strand cDNA was generated using random hexamer-primed reverse transcription, followed by second-strand cDNA synthesis. Then, A-Tailing Mix and RNA Index Adapters were added by incubation for end repair. The cDNA fragments obtained from the previous step were amplified by PCR, and the products were purified with Ampure XP Beads then dissolved in the EB solution. The product was validated with the Agilent Technologies 2100 Bioanalyzer for quality control. The double-stranded PCR products from the previous step were heat-denatured and circularized by the splint oligo sequence to produce the final library. The single-strand circular DNA (ssCir DNA) was formatted as the final library. The final library was amplified with phi29 to make DNA nanoballs (DNBs), which had more than 300 copies of one molecule. The DNBs were loaded into the patterned nanoarray and pair-end 100 base reads were generated on the BGIseq500 platform (BGI-Shenzhen, China). The measurements were conducted in triplicate.

### 4.4. Raw Data Analysis and Alignment of Reads to the Reference Genome

Clean reads were obtained by removing the adaptor sequences, and reads containing poly-N and low-quality reads were counted by using Soapnuke (v1.4.0, parameters: -l 5 -q 0.5 -n 0.1) and filtered by using Trimmomatic (v0.36, parameters: ILLUMINACLIP:2:30:10 LEADING:3 TRAILING:3 SLIDINGWINDOW:4:15 MINLEN:50). The filtered “clean reads” were saved in fastq format. Then, we compared the clean reads to the reference genome sequence by HISAT2 (v2.0.4) [73] and to the reference gene sequences by bowtie2 (V2.2.5) [74], and we calculated the expression level of the genes and transcripts by Rsem. To obtain the homologs of genes with known functions, the Basic Local Alignment Search Tool (BLAST) was utilized for sequence alignments using the TF (PlantTFDB), PRGdb (Plant Resistance Gene Database), GO (GeneOntology), KEGG (Kyoto Encyclopedia of Genes and Genomes) and Nr (NCBI non-redundant protein sequences) databases [75,76,77,78,79,80].

### 4.5. Identification of Differentially Expressed Genes

The FPKM method was used to eliminate the effect of different gene lengths and sequencing levels on the calculation of gene expression. Therefore, the FPKM values were used to compare gene expression differences between different samples. The DEGseq method is based on the Poisson distribution [33] (parameters: fold change ≥ 2 and adjusted *p*-value ≤ 0.001). *p*-values were corrected to Q-values by methods [81,82] and to improve the accuracy of the DEGs, we defined genes with multiples of differences more than twice and a Q-value ≤ 0.001, and screened them as significantly differentially expressed genes.

### 4.6. Quantitative Real-Time PCR

Total RNA from the seeds was isolated using the RNAprep Pure Plant Kit (Polysaccharides and Polyphenolics, Tiangen, Beijng, China) according to the manufacturer’s instructions. Three duplicates of total RNA extracts were reverse transcribed for first-strand cDNA synthesis for qRT-PCR using the TransScript^®^ All-in-One First-Strand cDNA Synthesis SuperMix for qPCR (One-Step gDNA Removal) (TransGen Biotech, Beijing, China). The qRT-PCR, performed by TransStart^®^ Tip Green qPCR SuperMix (TransGen Biotech, Beijing, China) with the total system was 20 μL, including 2 μL cDNA. The qRT-PCR conditions followed the manufacturer’s instructions. The primers were designed using Primer 6.0 software. The thermal cycling protocol was 30 s at 94 °C, then 40 cycles of 94 °C for 5 s and 60 °C for 30 s for annealing and extension. The specificity was assessed by the melt curve and size estimation of the amplified product. The expression levels of the DEGs were determined using the 2−∆∆t method. The reference large subunit ribosomal protein gene used for normalization of the assayed genes was Che034253 because its expression is stable throughout seed germination. The results shown are the average of three independent biological replicates, repeated three times.

## 5. Conclusions

In this study, the physiological data showed that starch accumulated briefly during germination, while soluble sugar continuously decreased before the leaves began to grow. Transcriptome analysis identified 15 categories and 49 functional genes were associated with starch and sucrose metabolism pathways, and the expression levels of the GBSS gene (Che029279) revealed that GBSS may be the key regulatory factor of starch synthesis. In addition, the transcriptome data suggested that β-amylase may be an important starch degrading enzyme and that SUSY mainly catalyzes the decomposition of sucrose during germination. By integrating RNA-Seq data with physiological and biochemical data, we have provided insights into the molecular mechanisms that regulate starch and sucrose metabolism in *C. henryi* seeds during germination. Most of the seeds of *Fagaceae* plants contain high starch content, and improving the understanding of starch metabolism during seed germination is helpful to understand the significance of starch to *Fagaceae* plants. Consequently, the genes associated with starch metabolism during germination will need to be more comprehensively characterized in future studies.

## Figures and Tables

**Figure 1 ijms-21-01431-f001:**
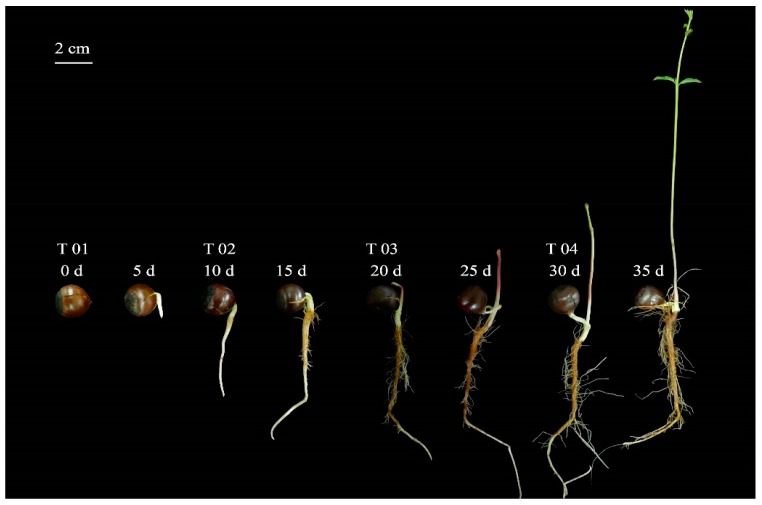
Eight seed germination stages of *Castanea henryi*. Four developmental stages were sampled for RNA-seq: T01, 0 days; T02, 10 days; T03, 20 days; T04, 30 days.

**Figure 2 ijms-21-01431-f002:**
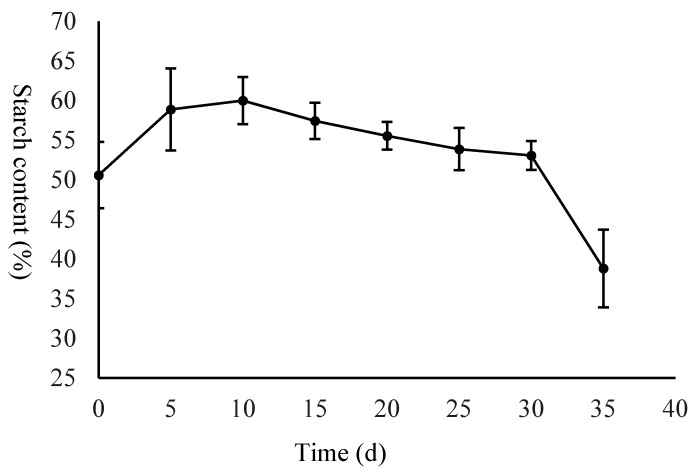
The changes in the starch content during germination. Values represent the X ± SD of three biological replicates.

**Figure 3 ijms-21-01431-f003:**
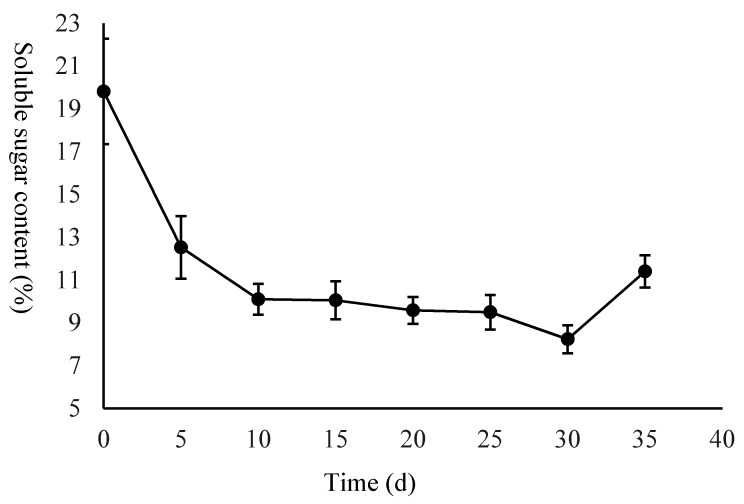
The changes in the soluble sugar content during germination. Values represent the mean X ± SD of three biological replicates.

**Figure 4 ijms-21-01431-f004:**
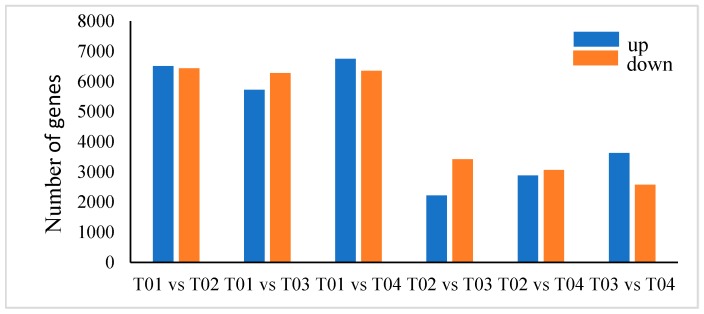
The numbers of up- and down-regulated genes among the four different stages.

**Figure 5 ijms-21-01431-f005:**
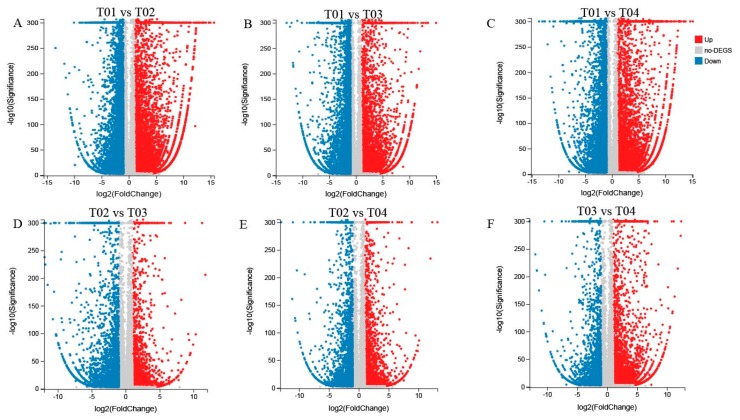
Volcano plots of the transcriptome between T01 and T02 (**A**), T01 and T03 (**B**), T01 and T04 (**C**), T02 and T03 (**D**), T02 and T04 (**E**), and T03 and T04 (**F**). The statistical significance (log10 of *p*-value; Y-axis) was plotted against log2-fold change (X-axis). The center of the volcano represents a fold change of zero, while the sides indicate downregulation (negative values) and upregulation (positive values).

**Figure 6 ijms-21-01431-f006:**
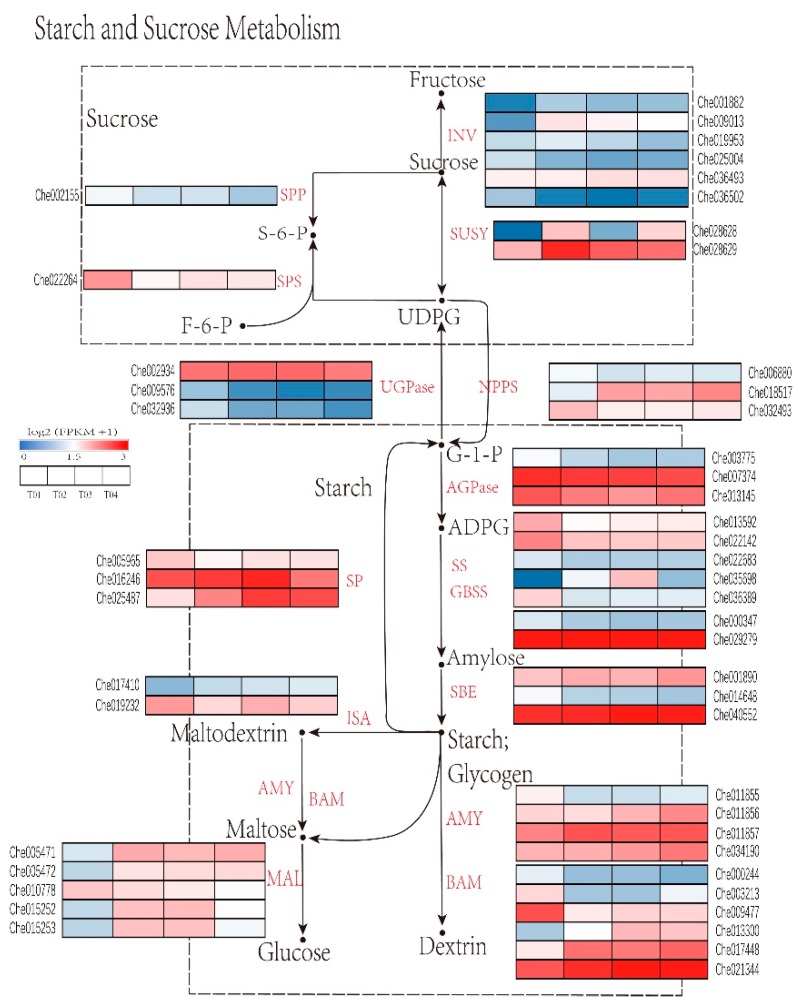
*C. henryi* genes that might be involved in the starch and sucrose metabolism pathways and their expression. The value of log2 (FPKM +1) is represented using the depth of color, with red representing the upregulated expression genes and blue representing the downregulated expression genes. FPKM means the fragments per kilobase of exon per million fragments mapped. INV—invertase; SuSy—sucrose synthase; SPS—sucrose–phosphate synthase; SPP—sucrose–phosphate phosphatase; UGPase—UDP-glucose pyrophosphorylase; NPPS—nucleotide pyrophosphatase; AGPase—ADP-glucose pyrophosphorylase; SS—starch synthases; GBSS—granule-bound starch synthase; SBE—starch-branching.

**Figure 7 ijms-21-01431-f007:**
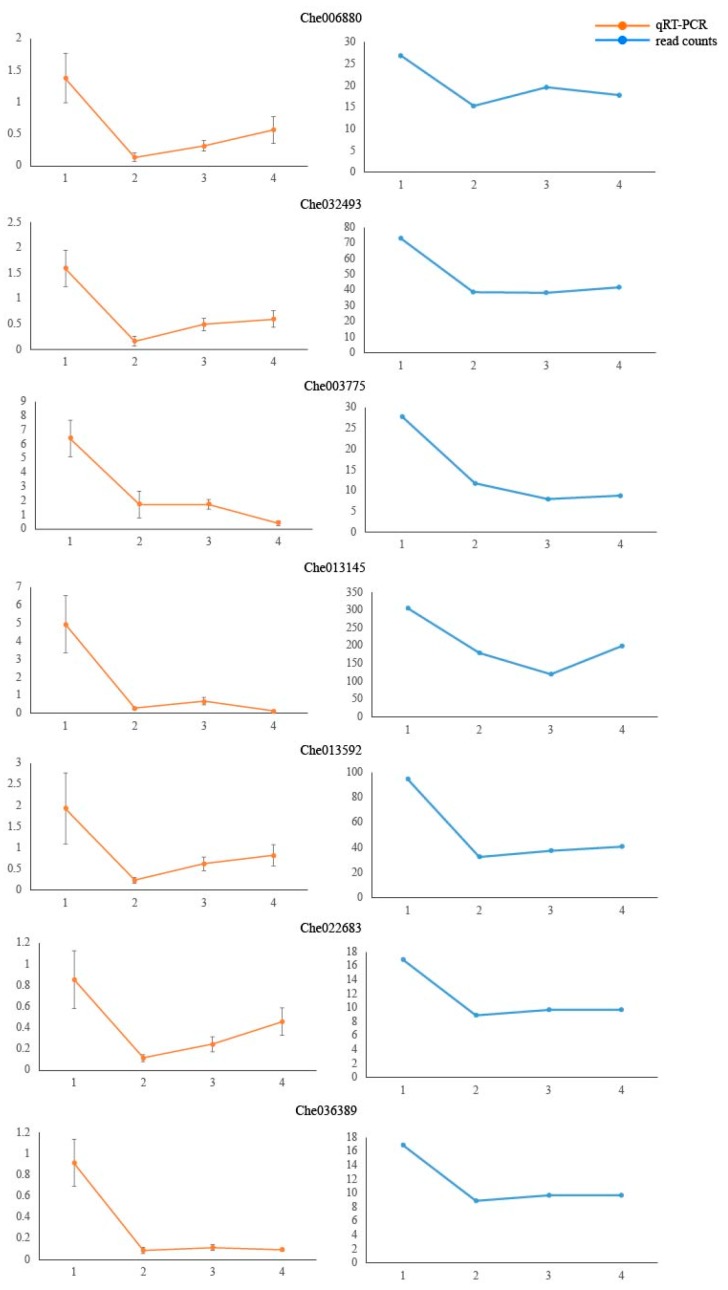
Quantitative real-time polymerase chain reaction analysis of the expression of starch synthesis-related genes. Analyses were completed in triplicate, and the error bars represent the standard errors.

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
