# Peer review of "Transcriptome Analysis and Identification of Genes Associated with Starch Metabolism in Castanea henryi Seed (Fagaceae)"

_ijms, 2020, doi:10.3390/ijms21041431_

Round 1

Reviewer 1 Report

The paper "Transcriptome analysis and identification of genes associated with starch metabolism in Castanea henryi seed (Fagaceae)" by Liu et al. is a very interesting work that deals with the analysis of sugar content in the germinating seeds of Henry chestnut. Transcriptomic analysis of the different enzymes putatively involved in starch and sugar metabolism is analyzed and related to the starch and soluble sugar content in the seeds. The work is correctly planned and the results are well presented. Nonetheless, there are several issues that prevent me from recommending its publication in its current form.

Please consider the following comments and suggestions, as well as I would appreciate if you could clear up some doubts.

- Which is the origin of the seeds? Are they all derived from the same crossing or genetically related?

- Line 452: How were the seeds dried (Temperature/time)?

- For measurements of sugar content or gene expression of stages other than the first one (day 0), What parts of the germinating seeds were used? Were radicles, etc..also used? This aspect should be clarified.

- Line 460: How do you "cool" samples to a constant volume?

- Line 462: the anthrone method should be further explained or referenced.

- Line 466: What are inclusion indicators?

- Line 471: Which manufacturer? Also in this line, to measure quality is not to qualify.

- Line 522-524: specificity was assessed with melt curve and size estimation. Where the authors keen to differentiate, for example, both NPSS genes (Che00688/Che03249) in an electrophoresis gel (232/224 bp)? Were the amplified products sequenced to confirm their identity?

- How much cDNA was used for qPCR reactions?

- How is it that the large subunit ribosomal protein was stable throughout  a metabolic demanding process such as germination? According to Supplementary Figure 5, there is a lot of activity within the translation-related genes.

- Line 61: "highest activity". What type of activity? Metabolic? Transcriptional?

- Line 103: "Castanea henryi is a member of the Fagaceae Castanea family." This sentence should be corrected.

- Figures 2 and 3: Starch content and soluble sugar contents are given as a percentage, but it is not specified over what total that percentage is (dry matter?). Besides, several figure legends (including these ones) do not give enough information.

- Line 162: Reads were mapped to the C. henryi genome. Is this genome available or has it been published? Besides, have the sequences from the transcriptomic experiment been archived in any public repository (SRA,...)?

- Line 173: 36,734 genes. Where does this data come from?

- Line 180: I think they were not predicted but rather assigned.

- Lines 263-286: though interesting, this part of the work should not be included in the Results section, it probably fits better in Discussion.

- Line 315: How could they be highest and then increase? Please clarify this issue.

- Line 317: "...decreased in T01 to T01".

- There is not statistical analysis of gene expression or sugar content.

- Hisat and Bowtie2 should be properly referenced.

- Line 507: this is not the proper way to reference someone´s work.

- Supplementary Figure 1: It is not the length of the transcriptomes, it is the length of the transcripts.

- Species names should always be in italics, also in the references.

- English grammar should be checked, as well as several typos could be found in the text. In this sense, a careful revision of the draft is needed.

To conclude, I would like to encourage the authors to take into consideration the comments above. They have developed a significant work with relevance for the scientific community, and the changes suggested might help improve the quality of the draft.

Author Response

Response to Reviewer 1 Comments

Point 1: Overall comments

The paper "Transcriptome analysis and identification of genes associated with starch metabolism in Castanea henryi seed (Fagaceae)" by Liu et al. is a very interesting work that deals with the analysis of sugar content in the germinating seeds of Henry chestnut. Transcriptomic analysis of the different enzymes putatively involved in starch and sugar metabolism is analyzed and related to the starch and soluble sugar content in the seeds. The work is correctly planned and the results are well presented. Nonetheless, there are several issues that prevent me from recommending its publication in its current form.

Please consider the following Points and suggestions, as well as I would appreciate if you could clear up some doubts.

Response 1: Thank you for your constructive suggestions; we have made revisions.

Point 2: Which is the origin of the seeds? Are they all derived from the same crossing or genetically related?

Response 2: Thank you for your patience. We have added a complete description in our manuscript according to your advice (line 475: The seeds which were genetically related were germinated).

Point 3: Line 452: How were the seeds dried (Temperature/time)?

Response 3: Thank you for your patience and suggestions. We have added a description in our manuscript according to your question (line 483: we dried the seeds at 50 ℃ for 72 hours to a constant).

Point 4: For measurements of sugar content or gene expression of stages other than the first one (day 0), What parts of the germinating seeds were used? Were radicles, etc..also used? This aspect should be clarified.

Response 3: Thank you for your patience and suggestions. We have added a description in our manuscript according to your question (line 484: including radicles, etc.).

Point 5: Line 460: How do you "cool" samples to a constant volume?

Response 5: Thank you for your patience and suggestions. We have added a description in our manuscript according to your question (line 463–464: kept it at a constant volume after room temperature cooling).

Point 6: Line 462: the anthrone method should be further explained or referenced.

Response 6: Thank you for your patience and suggestions. We have added a reference in our manuscript according to your advice (line 465–466: referencing the anthrone colorimetry method [72].).

Point 7: Line 466: What are inclusion indicators?

Response 7: Thank you for your patience. We have revised words in our manuscript according to your question (line 469–470: Based on the previous results of the physical indicators of and changes in soluble sugar and starch content).

Point 8: Line 471: Which manufacturer? Also in this line, to measure quality is not to qualify.

Response 8: Thank you for your patience and suggestions. We have added a complete description in our manuscript according to your question (line 474–488), and in addition, Nano Drop and Agilent 2100 Bioanalyzer (Thermo Fisher Scientific, MA, USA) can detect RNA and judge whether it meets the experimental standard.

Point 9: Line 522-524: specificity was assessed with melt curve and size estimation. Where the authors keen to differentiate, for example, both NPSS genes (Che00688/Che03249) in an electrophoresis gel (232/224 bp)? Were the amplified products sequenced to confirm their identity?

Response 9: Thank you for your patience and suggestions. The amplification efficiency of the objective gene primers is qualified, which can completely explain the specificity of the primers, so the amplification products have not been sequenced.

Point 10: How much cDNA was used for qPCR reactions?

Respond 10: Thank you for your patience and suggestions. We have added a complete description in our manuscript according to your question (lines 540: including 2 μL cDNA.).

Point 11: How is it that the large subunit ribosomal protein was stable throughout a metabolic demanding process such as germination? According to Supplementary Figure 5, there is a lot of activity within the translation-related genes.

Respond 11: Thank you for your patience and suggestions. The large subunit ribosomal protein is a relatively stable housekeeping gene protein, and through the analysis of the difference of transcriptome data, it is concluded that the gene is the most stable.

Point 12: Line 61: "highest activity". What type of activity? Metabolic? Transcriptional?

Respond 12: Thank you for your patience and suggestions. We have added a description in our manuscript according to your question (line 61: highest metabolic activity).

Point 13: Line 103: "Castanea henryi is a member of the Fagaceae Castanea family." This sentence should be corrected.

Respond 13: Thank you for your patience and suggestions. We revised this sentence according to your advice (line 103: Castanea henryi is a member of the Castanea of Fagaceae family).

Point 14: Figures 2 and 3: Starch content and soluble sugar contents are given as a percentage, but it is not specified over what total that percentage is (dry matter?). Besides, several figure legends (including these ones) do not give enough information.

Respond 14: Thank you for your patience and suggestions. We have added a description in our manuscript according to your question (line 137: the dry matter).

Point 15: Line 162: Reads were mapped to the C. henryi genome. Is this genome available or has it been published? Besides, have the sequences from the transcriptomic experiment been archived in any public repository (SRA,..)?

Respond 15: Thank you for your patience and suggestions. At present, C. henryi genome data have not been published and transcriptomic experiment data have not been uploaded.

Point 16: Line 173: 36,734 genes. Where does this data come from?

Respond 16: Thank you for your patience and suggestions. We got the data according to the results of the annotation, and the details can be referred to in Supplementary Table. 3.

Point 17: Line 180: I think they were not predicted but rather assigned.

Respond 17: Thank you for your patience and suggestions. We revised words according to your advice (line 180: assign).

Point 18: Lines 263–286: though interesting, this part of the work should not be included in the Results section, it probably fits better in Discussion.

Respond 18: Thank you for your patience and suggestions. This part briefly introduces the importance and connection of the selected key genes. Keeping this part makes the structure of the article complete and the connection compact. In addition, this part is introduced in detail in the discussion. Therefore, we didn't put this part into the discussion.

Point 19: Line 315: How could they be highest and then increase? Please clarify this issue.

Respond 19: Thank you for your patience and suggestions. We have added a clear description in our manuscript according to your question (line 316: decreased significantly in T01–T02, and increased slightly in T02–T04.).

Point 20: Line 317: "...decreased in T01 to T01".

Respond 20: Thank you for your patience and suggestions. We have added a correct description in our manuscript according to your question (line 318: in T01 to T03).

Point 21: There is not statistical analysis of gene expression or sugar content.

Respond 21: Thank you for your patience and suggestions. The focus of this paper is to explore the key genes of starch metabolism; the main content is included in the part describing gene selection, so the statistical analysis is not included.

Point 22: Hisat and Bowtie2 should be properly referenced.

Respond 22: Thank you for your patience and suggestions. We have added a reference in our manuscript according to your advice (line 514: HISAT2 (v2.0.4) [73] and to the reference gene sequences by bowtie2 (V2.2.5) [74]).

Point 23: Line 507: this is not the proper way to reference someone´s work.

Respond 23: Thank you for your patience and suggestions. We have revised the way of referencing in our manuscript according to your advice (line 527–528: P-values were corrected to Q-values by methods [81–82]).

Point 24: Supplementary Figure 1: It is not the length of the transcriptomes, it is the length of the transcripts.

Respond 24: Thank you for your patience and suggestions. We have revised the words in our manuscript according to your advice.

Point 25: Species names should always be in italics, also in the references.

Respond 25: Thank you for your patience and suggestions. We have modified the font format of the species name in the manuscript according to your suggestion.

Point 26: English grammar should be checked, as well as several typos could be found in the text. In this sense, a careful revision of the draft is needed.

Respond 26: Thank you for your patience. We have checked the English grammar and revised the draft carefully.

Reviewer 2 Report

Overall comments:

The Objective of the study is to identify the starch metabolizing encoding genes in the germinating seeds of Castanea henryi using RNA-seq transcriptomic analysis. Genes encoding starch metabolizing genes were identified using homologous search against KEGG databases. Candidate genes were validated using real time quantitative PCR. Along with RNA-seq analysis, starch content and reducing sugar content was also measured to identify the germination stages used for transcriptomic analysis.

Strengths: Authors did a thorough job in describing the starch metabolizing pathways and identifying the genes participating in the metabolic pathway.

Weakness: Unfortunately, there are more weaknesses in the paper and therefore will not be accepted at this stage for publication. There are inconsistencies in the introduction, results and misrepresentation of the facts. Manuscript is written very vaguely and specifics needs to be included wherever appropriate. Some examples are provided in the specific comments. The writing also needs to be improved and checked for spelling mistakes. Also, there are many typos throughout the manuscript. Please find the specific comments below.

Line 31: Mention the rationale for studying starch metabolism in this species

Line 39: Be more specific in the intensity of changes observed. For example rather than using “first increased than decreased”, use specific hours in which these changes were observed.

Line 43: Change Syntheses to synthesis

Line 61: Highest activity of?

Line 64: Which substances?

Line 75: “Series”

Line 84: Sucrose synthase is involved in the synthesis of sucrose not hydrolysis. Please check the information in the introduction.

Line 142: Orientation of the x-axes title is reversed

Line 149: Based on the starch content analysis, there is no significant differences in the starch content during the time points selected for transcriptional analysis. Rather a comparison of Time 0, 20 and 35 days would have been more informative. Please explain the rationale for your time points in detail.

Line 174: Mention the databases in this line

Line 172: What is the rationale of identifying the genes homologous to resistance genes as the objective of the study is to identify the starch metabolism genes?

Line 232: Figure 6 needs to be increased in resolution.

Line 257: There is inconsistency with the figure and this sentence.

Line 274: Figure shows only 2 ISAs?

Line 312: Please mention the gene annotation also along with the contig name

Line 364 and 368: missing citation

Line 437: How would studying the starch metabolism in this plant will help in understanding the role of Fagaceae plants? What role do Fagaceae plants play? This sentence is very confusing

Line 471: Mention the manufacturers

Line 480: Which kit was used for cDNA library preparation?

Line 489: Versions of the packages, parameters used and the references are missing.

Line 506: Is this supposed to be a reference?

Line 516: Be consistent with either using RT-qPCR or qRT-PCR

Line 525: How was this tested?

Line 531: Supplementary material information is not included.

Author Response

Response to Reviewer 2 Comments

Point 1: Overall Comments

The Objective of the study is to identify the starch metabolizing encoding genes in the germinating seeds of Castanea henryi using RNA-seq transcriptomic analysis. Genes encoding starch metabolizing genes were identified using homologous search against KEGG databases. Candidate genes were validated using real time quantitative PCR. Along with RNA-seq analysis, starch content and reducing sugar content was also measured to identify the germination stages used for transcriptomic analysis.

Strengths: Authors did a thorough job in describing the starch metabolizing pathways and identifying the genes participating in the metabolic pathway.
Weakness: Unfortunately, there are more weaknesses in the paper and therefore will not be accepted at this stage for publication. There are inconsistencies in the introduction, results and misrepresentation of the facts. Manuscript is written very vaguely and specifics needs to be included wherever appropriate. Some examples are provided in the specific Points. The writing also needs to be improved and checked for spelling mistakes. Also, there are many typos throughout the manuscript. Please find the specific Points below.

Response 1: Thank you for your constructive suggestions, we have made revisions to the introduction, results and misrepresentation of the facts based on your Points. Please see the introduction, results and related parts.

Point 2: Line 31: Mention the rationale for studying starch metabolism in this species

Response 2: Thank you for your patience and suggestions. We revised this sentence according to your advice (line 31–32: Starch is the most important form of carbohydrate storage and is the major energy reserve in some seeds, especially Castanea henryi).

Point 3: Line 39: Be more specific in the intensity of changes observed. For example rather than using “first increased than decreased”, use specific hours in which these changes were observed.

Response 3: Thank you for your patience and suggestions. We have revised the description of the results in our manuscript according to your advice (line 38–41: The results showed that the starch content increased in 0–10 days and decreased in 10–35 days, while the soluble sugar content continuously decreased in 0–30 days and increased in 30–35 days).

Point 4: Line 43: Change Syntheses to synthesis

Response 4: Thank you for your patience and suggestions. We have revised the word in our manuscript according to your suggestion (line 43: starch synthases).

Point 5: Line 61: Highest activity of?

Response 5: Thank you for your patience and suggestions. We have added a description in our manuscript according to your question (line 61: highest metabolic activity).

Point 6: Line 64: Which substances?

Response 6: Thank you for your patience and suggestions. We revised this sentence according to your advice (line 64: proteins and carbohydrates provide energy).

Point 7: Line 75: “Series”

Response 7: Thank you for your patience and suggestions. We have revised the word in our manuscript according to your suggestion (line 75: the amount of).

Point 8: Line 84: Sucrose synthase is involved in the synthesis of sucrose not hydrolysis. Please check the information in the introduction.

Response 8: Thank you for your patience and suggestions. Invertible reaction catalyzed by sucrose synthase can promote the synthesis and degradation of sucrose, which is supported by our references.

Point 9: Line 142: Orientation of the x-axes title is reversed

Response 9: Thank you for your patience and suggestions. We have changed the Orientation of the x-axes title according to your question.

Point 10: Line 149: Based on the starch content analysis, there is no significant differences in the starch content during the time points selected for transcriptional analysis. Rather a comparison of Time 0, 20 and 35 days would have been more informative. Please explain the ationale for your time points in detail.

Response 10: Thank you for your patience and suggestions. The focus is on the change in trend of starch content of this paper; the timepoint selected in this paper can better reflect the change trend in starch content, first increasing and then decreasing.

Point 11: Line 174: Mention the databases in this line

Response 11: Thank you for your patience and suggestions. ‘Database’ refers to the five public databases used for annotation in this article. Information can be found in Supplementary Table. 3.

Point 12: Line 172: What is the rationale of identifying the genes homologous to resistance genes as the objective of the study is to identify the starch metabolism genes?

Response 12: Thank you for your patience and suggestions. The identification of genes homologous with disease resistance genes is to reveal more gene information.

Point 13: Line 232: Figure 6 needs to be increased in resolution.

Response 13: Thank you for your patience and suggestions. We have increased the resolution of Figure 6 according to your advice.

Point 14: Line 257: There is inconsistency with the figure and this sentence.

Response 14: Thank you for your patience and suggestions. The figure is consistent with the original content; red means high expression, blue means low expression, white is transition, so che006880 and che032493 are the highest expression in T01 period.

Point 15: Line 274: Figure shows only 2 ISAs?

Response 15: Thank you for your patience and suggestions. We have added a correct description in our manuscript according to your question (line 274: two ISA).

Point 16: Line 312: Please mention the gene annotation also along with the contig name

Response 16: Thank you for your patience and suggestions. We have added clear description in our manuscript according to your question (line 312–319: while the transcript of Che006880 (NPPS) decreased significantly in T01 to T02, gradually increased in T02 to T03, and then gradually decreased in T03 to T04. Moreover, the expression levels of Che032493 (NPPS), Che013592 (SS), Che022683 (SS), and Che036389 (SS) were all highest in T01, decreased significantly in T01–T02, and increased slightly in T02–T04. Furthermore, the expression levels of Che003775 (AGPase) and Che013145 (AGPase) were significantly decreased in T01 to T03, but then the expression of Che00377 (AGPase) slightly increased, and the expression of Che013145 (AGPase) significantly increased).

Point 17: Line 364 and 368: missing citation

Response 17: Thank you for your patience and suggestions. The citation already exists in the text (line 369: [61]).

Point 18: Line 437: How would studying the starch metabolism in this plant will help in understanding the role of Fagaceae plants? What role do Fagaceae plants play? This sentence is very confusing

Response 18: Thank you for your patience and suggestions. We have added a clear description in our manuscript according to your question (line 438–440: Most of the seeds of Fagaceae plants contain high starch content, and improving the understanding of starch metabolism during seed germination is helpful to understand the significance of starch to Fagaceae plants.).

Point 19: Line 471: Mention the manufacturers

Response 19: Thank you for your patience and suggestions. We have added a complete description in our manuscript according to your question (line 474–488).

Point 20: Line 480: Which kit was used for cDNA library preparation?

Response 20: Thank you for your question. The kit used is independently developed by BGI company with a pending patent and will not be released for now.

Point 21: Line 489: Versions of the packages, parameters used and the references are missing.

Response 21: Thank you for your patience and suggestions. We have added the versions of the packages, parameters used and the references in our manuscript according to your advice (line 509–514: Soapnuke (v1.4.0, parameters: -l 5 -q 0.5 -n 0.1) and filtered by using Trimmomatic (v0.36, parameters: ILLUMINACLIP:2:30:10 LEADING:3 TRAILING:3 SLIDINGWINDOW:4:15 MINLEN:50). The filtered "clean reads" were saved in fastq format. Then, we compared the clean reads to the reference genome sequence by HISAT2 (v2.0.4) [73] and to the reference gene sequences by bowtie2 (V2.2.5) [74]).

Point 22: Line 506: Is this supposed to be a reference?

Response 22: Thank you for your patience and suggestions. We have revised the way of referencing in our manuscript according to your advice (line 527-528: P-values were corrected to Q-values by methods [81–82]).

Point 23: Line 516: Be consistent with either using RT-qPCR or qRT-PCR

Response 23: I Thank you for your patience and suggestions. We have revised the description in our manuscript according to your advice (line 306–310).

Point 24: Line 525: How was this tested?

Response 24: Thank you for your patience and suggestions. The large subunit ribosomal protein is a relatively stable housekeeping gene protein, and through the analysis of the difference of transcriptome data, it was concluded that the gene is the most stable.

Point 25: Line 531: Supplementary material information is not included.

Response 25: Thank you for your patience and suggestions. We have eliminated this part according to your advice.

Round 2

Reviewer 1 Report

Minor comments:

  • Line 47: "affect".
  • Line 103: "Castanea genus of the Fagaceae family".
  • Line 109: "flour". What flour?
  • LIne 328: "rich in nutritional elements".
  • Line 407: "must be catalysed".
  • Line 623: "Handbook".
  • Line 779: "Altschul".
  • Supp. Fig. 1: length of transcripts.

Author Response

Response to Reviewer 1 Comments

Point 1: Line 47: "affect".

Response 1: Thank you for your patience and suggestions. We have revised the word in our manuscript according to your suggestion (line 47: affect).

Point 2: Line 103: "Castanea genus of the Fagaceae family".

Response 2: Thank you for your patience and suggestions. We have revised the word in our manuscript according to your suggestion (line 103: Castanea genus of the Fagaceae family).

Point 3: Line 109: "flour". What flour?

Response 3: Thank you for your patience and suggestions, flour is ground from wheat.

Point 4: Line 328: "rich in nutritional elements".

Response 4: Thank you for your patience and suggestions. We have revised the word in our manuscript according to your suggestion (line 328–329: rich in nutritional elements).

Point 5: Line 407: "must be catalysed".

Response 5: Thank you for your patience and suggestions. We have revised the word in our manuscript according to your suggestion (line 407: must be catalysed).

Point 6: Line 623: "Handbook".

Response 6: Thank you for your patience and suggestions. We have revised the word in our manuscript according to your suggestion (line 623: Handbook).

Point 7: Line 779: "Altschul".

Response 7: Thank you for your patience and suggestions. We have revised the word in our manuscript according to your suggestion (line 779: Altschul).

Point 8: Supp. Fig. 1: length of transcripts.

Response 8: Thank you for your patience and suggestions. We have revised the word in our manuscript according to your suggestion. (Supp. Fig. 1: length of transcripts.)
